# ContraDiff: Unifying Training Process of Generative and Discriminative Vision Tasks in One Diffusion Model

## Abstract

Besides unprecedented ability in image generation, text-to-image diffusion models are also able to provide powerful intermediate representations that support various discriminative vision tasks. However, efficiently adapting these models to handle both generative and discriminative tasks remains largely unexplored. While some unified frameworks have been proposed to reduce the overhead of training pipelines, they often rely on computationally expensive pretraining processes and lack flexibility in adaptation. In this paper, we propose ContraDiff, a novel framework to efficiently leverage a pretrained diffusion model for both generative and discriminative tasks. Our approach focuses on unified training and parameter-efficient optimization. Our framework combines a reconstruction loss and a contrastive loss on images with varying noise levels to effectively balance generative and contrastive training. Additionally, we apply LoRA to a pre-trained Stable Diffusion model, reducing the number of trainable parameters and the optimization overhead without compromising performance. Our experiments show that ContraDiff excels in both generative and discriminative vision tasks. Our model achieves 80.1% accuracy on ImageNet-1K classification and an FID of 5.56 for ImageNet 256×256 unconditional image generation, all while requiring significantly fewer trainable parameters. This efficiency offers advantages in computational resources and enhances the model's adaptability across a range of vision tasks. The code will be released publicly upon acceptance.

## 1 Introduction

Self-supervised representation learning has demonstrated remarkable results in deriving rich, transferable features without additional supervision signals. Contrastive approaches (Chen et al., 2020; Caron et al., 2021) and generative methods (He et al., 2022; Xie et al., 2022) have been developed along separate paths to learn robust visual representations. However, recent research (Mukhopadhyay et al., 2023; Park et al., 2023) suggests that both contrastive and generative paradigms have shared underlying principles in capturing semantic information from unlabeled data.

Following this idea, several methods (Li et al., 2023b; Hudson et al., 2024; Zhu et al., 2024) have aimed to unify self-supervised learning for both generative and discriminative tasks. However, these methods still encounter notable limitations, particularly in balancing the trade-off between feature robustness for recognition and high-quality generation (He et al., 2022). For example, SODA (Hudson et al., 2024) jointly trains a conditional encoder and a denoising network, introducing a compact modulation in the encoder's latent space. However, as SODA is trained from scratch, it does not fully exploit the generative capabilities of pretrained diffusion models. Another challenge arises largely from the extensive computational demands. A state-of-the-art MAGE model (Li et al., 2023b), for example, relies on a heavy-parameterized ViT-L/16 backbone with over 400M trainable parameters, requiring 1600 epochs of training. This high resource demand limits the practicality of such models in real-world applications. This raises a critical research question in self-supervised representation learning: **Can we develop a unified framework that effectively balances feature robustness and generation quality while being computationally efficient?**

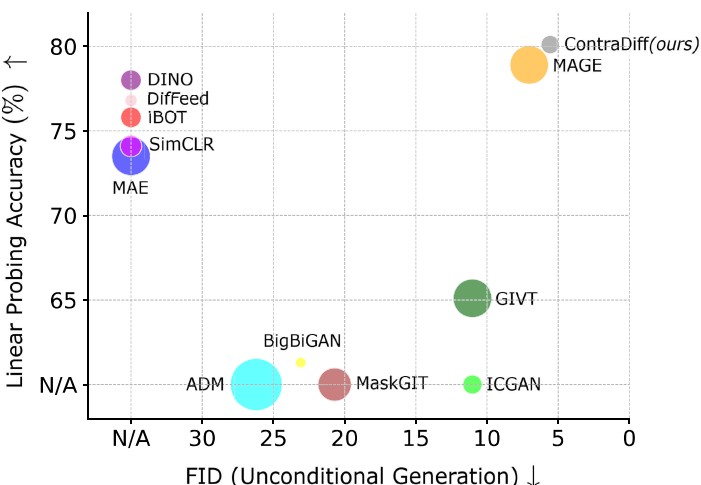

Figure 1: **ContraDiff balances accuracy and efficiency.** We report linear probing and unconditional image generation performance of different methods on ImageNet-1K. The area of a circle corresponds to the number of trainable parameters. Our method outperforms baseline models in both discriminative (classification) and generative (unconditional image generation) tasks, even surpassing those trained for only one of these tasks. In the meantime, our method maintains a small number of trainable parameters to reduce training resource overhead.

Remarkable advancements in generative models present a promising direction for the question. Diffusion models, in particular, have emerged as a powerful framework for high-fidelity image generation (Ho et al., 2020) and meaningful representation learning (Preechakul et al., 2022; Mittal et al., 2023), suggesting a unique opportunity to unify generative and discriminative tasks under a single framework. Specifically, Stable Diffusion (SD) is a powerful pre-trained latent diffusion model (LDM) (Rombach et al., 2022) that generates images by progressively denoising random Gaussian noise into coherent visuals through a UNet architecture (Ronneberger et al., 2015). Unlike traditional diffusion models operating in pixel space, it functions in a lower-dimensional latent space, enabling efficient processing and generation. Modern Stable Diffusion models are pre-trained on large scale datasets (Schuhmann et al., 2022) and open-sourced, making fine-tuning and fast adaptation on them efficient without the need for training from scratch.

In this work, we propose CONTRADIFF, a novel unified framework that integrates representation learning and generative modeling within a single diffusion process. Our key technical novelty is the incorporation of contrastive learning into diffusion models - In the reverse diffusion process, where images are progressively denoised through sequential steps, contrastive loss can be naturally applied by treating images at different noise levels as distinct "views" of the same underlying data. Inspired by SimCLR (Chen et al., 2020), we incorporate a contrastive objective that operates across varying noise levels, leveraging both the efficiency and discriminative benefits of contrastive learning. This enables ContraDiff to learn robust features for discriminative tasks while preserving its ability to generate high-fidelity images.

To address the high computational demands inherent to large-scale diffusion models, we integrate LoRA (Hu et al., 2021) as an efficient adaptation mechanism. LoRA enables the insertion of low-rank, learnable updates directly into the weight matrices, eliminating the need for full model fine-tuning. Specifically, we apply LoRA to the cross-attention matrices in Stable Diffusion during training, enabling efficient UNet weight updates that align with the image condition latent at minimal computational costs. By reducing resource requirements, LoRA-enhanced diffusion allows for the simultaneous application of representation learning and generative modeling within a unified framework, reducing adaptation cost while preserving generative quality.

Our framework demonstrates competitive classification accuracy and high-quality image generation on ImageNet-1K (Russakovsky et al., 2015), outperforming certain task-specific contrastive methods. Through comprehensive empirical evaluation, we highlight the effectiveness of unifying contrastive and generative learning, showing that these approaches can coexist within a single framework to yield strong results across both classification and image synthesis tasks. In summary, our main contributions are as follows:

i) **A unified framework** that bridges representation learning and generative modeling by learning contrastive features obtained from generative denoising steps in diffusion processes, boosting both image generation and classification performance.

ii) **Comprehensive empirical evaluation** on ImageNet-1K, demonstrating strong image generation capabilities alongside high classification accuracy. Additionally, transfer learning experiments on CIFAR-100 confirm the generalization ability of our method.

## 2 Related Work

**Self-supervised learning in recognition tasks.** Self-supervised learning has transformed computer vision by enabling models to learn from unlabeled data using its inherent structure to create supervision signals. Early advances in the area were driven by contrastive methods, where models learn meaningful representations by contrasting positive and negative sample pairs. Pioneering methods like SimCLR (Chen et al., 2020) and MoCo (He et al., 2020) maximize similarity between different views of the same image, contrasting these with other images. Later, non-contrastive approaches such as DINO (Caron et al., 2021) introduced a teacher-student self-distillation approach, where the student model learns to match representations from a teacher network. Collectively, these methods have shown that contrastive and distillation-based self-supervision can learn high-quality representations without labeled data. Recent works have also applied self-supervised learning to generative modeling. Masked autoencoders (MAE) (He et al., 2022) demonstrated that transformer-based architectures can learn strong visual representations by learning to reconstruct masked image regions.

However, most early self-supervised learning methods require extensive pretraining to reach competitive performance, often training from scratch over hundreds of epochs. For example, DINO (Caron et al., 2021) achieves optimal performance after training for 800 epochs, demanding substantial computational resources. This prolonged training time limits the practical scalability of self-supervised learning, particularly for research and applications constrained by resources. Furthermore, while generative models like MAE (He et al., 2022) demonstrate promising reconstruction abilities, they often struggle to balance image fidelity with robust feature learning, especially in high-fidelity generative tasks. Consequently, there is a pressing need for methods that unify robust feature extraction with high-quality generation within a more resource-efficient, self-supervised framework.

**Diffusion model for discriminative tasks.** Diffusion models (Sohl-Dickstein et al., 2015; Ho et al., 2020) are a class of generative models that progressively convert random noise into high-fidelity image samples. In addition to recent works (Saharia et al., 2022; Ramesh et al., 2022; Rombach et al., 2022) that achieved remarkable results in high-quality and diverse image synthesis, their potential for representation learning has gained attention due to their ability to capture rich, hierarchical features. Several adaptations target discriminative tasks: DiffAE (Preechakul et al., 2022) uses an auto-encoding process within the diffusion framework, effectively reconstructing input data from noise to capture meaningful latent features. DiffMAE (Wei et al., 2023) combines diffusion with masked autoencoders, enhancing feature extraction and generalization by reconstructing partially corrupted inputs. Diffusion Classifier (Li et al., 2023a) further extends diffusion models to classification tasks by training the model to recognize classes within noisy data, enhancing its discriminative capabilities.

However, adapting diffusion models for self-supervised learning still presents challenges. These models are inherently large (Rombach et al., 2022; Karras et al., 2022), making full fine-tuning computationally expensive. Additionally, current approaches to feature extraction, such as DiffFeed (Mukhopadhyay et al., 2023) and DDAE (Xiang et al., 2023), often depend on frozen, pretrained diffusion models. This limits flexibility

when extending to other discriminative tasks, as frozen models may not adapt effectively across different contexts. For example, DiffFeed (Mukhopadhyay et al., 2023) uses multiple features from a fixed guided diffusion model (Dhariwal & Nichol, 2021) to explore feature fusion strategies, which is model-specific and does not improve the model's generative capability.

**Unified self-supervised learning for discriminative and generative tasks.** Recent advancements in unified self-supervised learning frameworks aim to support both discriminative and generative tasks within a single model, reflecting a shift towards versatile, efficient learning paradigms. MAGE (Li et al., 2023b) introduces a self-supervised approach that learns joint representations for both tasks via a novel masking strategy and a contrastive loss. However, MAGE requires an extensive pretraining phase to achieve robust representations, making it resource-intensive. For diffusion models, SODA (Hudson et al., 2024) employs a compact bottleneck to the representation from its DDPM (Ho et al., 2020) conditional encoder, training separate encoder and generator modules for unified task execution. Despite these advances, existing frameworks often depend on heavy pretraining and substantial computational resources, which limit their adaptability. This underscores the need for a resource-efficient unified framework capable of high performance in both discriminative and generative tasks with minimal computational overhead.

## 3 Method

### 3.1 Preliminaries for diffusion models

Diffusion models have emerged as a powerful class of generative models, known for their ability to generate high-quality images by modeling the data generation process as a reverse diffusion process.

**Forward process.** A diffusion model operates through a sequence of gradual, noise-adding transformations that convert data from a complex distribution into a simpler distribution (e.g., a Gaussian distribution) over a predefined number of steps. This process is inspired by non-equilibrium thermodynamics (Sohl-Dickstein et al., 2015) and has been refined across the works of Song et al. (2020b); Ho et al. (2020); Song et al. (2020a). Formally, the diffusion forward process can be described by a discrete Markov chain in Equation (1), where $x_t$ represents noisy data at discrete time step $t$, $\beta_t$ is the variance schedule which controls the noise level at each step, progressively transforming the data into noise.

$$q(x_t|x_{t-1}) = \mathcal{N}\left(\sqrt{1 - \beta_t}\, x_{t-1}, \beta_t I\right) \tag{1}$$

**Reverse process.** The reverse process, which is the core of a diffusion model's generative capability, aims to reconstruct the original data distribution $x_0 \sim p_{\text{data}}(x)$ from the noise. The DDPM reverse process is formalized as Equation (2), where $\alpha_t := 1 - \beta_t$, $\bar{\alpha}_t := \prod_{s=1}^{t} \alpha_s$, $\epsilon \sim \mathcal{N}(0, I)$, and $\epsilon_\theta(x_t, t)$ is a neural network that learns to predict the noise component with $x_t$ and $t$.

$$x_{t-1} = \frac{1}{\sqrt{1 - \beta_t}}\left(x_t - \frac{\beta_t}{\sqrt{1 - \bar{\alpha}_t}}\epsilon_\theta(x_t, t)\right) + \sqrt{\frac{1 - \bar{\alpha}_{t-1}}{1 - \bar{\alpha}_t}\beta_t} \cdot \epsilon \tag{2}$$

**Latent diffusion models (LDM).** During training, LDMs first compress input images into a low-dimensional latent $z$ with a pre-trained visual encoder $\mathcal{E}$, then perform noise-adding and denoising in latent space, and decode reconstructed latent via a decoder $\mathcal{D} : \tilde{x} = \mathcal{D}(\tilde{z})$, where $z = \mathcal{E}(x)$. This compression procedure preserves semantic information of image data while being more efficient in terms of computational resources, as evidenced by Rombach et al. (2022).

### 3.2 Method Overview

CONTRADIFF extends the capabilities of a pre-trained Stable Diffusion model beyond generative tasks through efficient fine-tuning and feature extraction for representation learning. As shown in Figure 2, the input image $x$ is first encoded into latent representations $z$ by a VAE latent encoder. An image conditioner

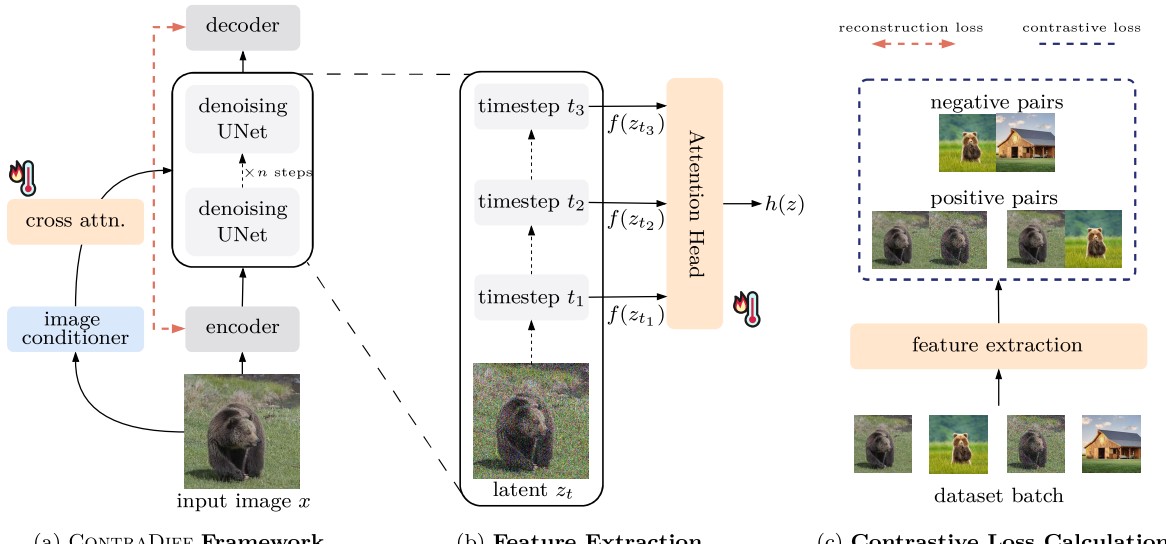

(a) CONTRADIFF **Framework**      (b) **Feature Extraction**      (c) **Contrastive Loss Calculation**

Figure 2: **Overview of the ContraDiff training pipeline.** An input image is encoded by a VAE encoder to produce a latent representation $z$, which is then perturbed with noise to form a noisy latent of level $t$. This noisy latent is processed by a denoising UNet with the conditional latent applied on cross-attention layers for $n$ steps. To enhance efficiency, LoRA is applied in the QKV (query, key, value) attention layers. This setup allows ContraDiff to balance generative and discriminative tasks effectively while reducing training resource requirements. The output of UNet is then decoded by a VAE decoder, reconstructing the image from the latent representation.

also generates image-based conditional latent using $x$. Next, Gaussian noise of level $t$ is added to the image latent following Equation (1), forming a noisy latent representation. The noisy latent, along with image condition embeddings, is then fed into the denoising UNet of the Stable Diffusion model, which reconstructs the latent representation before it is decoded back into pixel space.

To achieve efficient adaptation while preserving the pre-trained weights, we integrate Low-Rank Adaptation (LoRA) matrices (Hu et al., 2021) within the cross-attention layers of the denoising UNet. This strategy facilitates flexible fine-tuning and enhances representation learning without incurring extensive computational costs. Detailed explanations of each component follow below.

### 3.3 Training Objectives

**Generative training.** For each input image $x$, we encode it into latent space: $z = \mathcal{E}(x)$. To retain the model's generative capabilities while adapting it to new tasks, we employ a reconstruction loss on the model's denoising output, following the LDM loss formulation. Our primary goal is learning to reconstruct noisy latent $z_t$, which is equivalent to predicting the noise added on image latent representations, as formulated in Equation (3):

$$\mathcal{L}_{\text{recon}} = \mathbb{E}_{x \sim p_{\text{data}}, \epsilon \sim \mathcal{N}(0, I), t} \left[ \| \epsilon - \epsilon_\theta \left( z_t, t \right) \|^2 \right],\tag{3}$$

where $x$ is the input image, $\epsilon$ represents noise sampled from a Gaussian distribution, and $z_t$ is the noisy latent, which can be obtained from the model by the forward process.

**Contrastive feature extraction.** We leverage the rich representations within the diffusion model by extracting features from the bottleneck layer of the UNet architecture, where spatial resolution is minimized, and semantic information is densely encoded. Specifically, during a denoising step $t$, when an image latent $z$ is passed through the UNet $\epsilon_\theta \left( z_t, t \right)$, we use the activation $f(z_t)$ from the bottleneck layer as the feature. To

further enhance the extracted features, we apply a cross-attention mechanism (Vaswani, 2017) to the output of the bottleneck layer at different denoising timesteps $(t_1, t_2, t_3)$:

$$h(z) = \texttt{Attention}\left[W_Q f(z_{t_1}), W_K f(z_{t_2}), W_V f(z_{t_3})\right] \tag{4}$$

where $W_Q$, $W_K$, and $W_V$ are learnable projection matrices for query, key, and value transformations, respectively. This strategy encodes features from different denoising steps, resulting in a representation enriched with consistent semantic information.

**Contrastive loss design.** Following the approach in Li et al. (2023b), we apply a contrastive learning strategy to enhance the separability of diffusion features for improved performance on discriminative tasks. To construct positive/ negative pairs, we treat different noise levels as unique "views" of an image. Specifically, given a clean image $x$, we first encode it into latent $z$. Then, we generate two distinct "views" of $z$ by applying different noise levels in the forward diffusion process:

$$z_t \sim q(z_t|z), z_{t'} \sim q(z_{t'}|z) \tag{5}$$

where $(t, t')$ are time steps sampled from a fixed schedule. We employ InfoNCE loss (Oord et al., 2018) to maximize the mutual information between features extracted from these noisy views:

$$\mathcal{L}_{\text{contrast}} = -\sum_{i=1}^{N} \log \frac{2 \cdot \exp(\text{sim}(h_i, h_i')/\tau)}{\sum_{j=1}^{N} \mathbb{1}_{j \neq i} \left(\exp(\text{sim}(h_i, h_j)/\tau) + \exp(\text{sim}(h_i, h_j')/\tau)\right)} \tag{6}$$

where $N$ is the batch size, $\text{sim}(\cdot, \cdot)$ represents cosine similarity, $\tau$ is a temperature parameter, $h_i$ represents feature extracted from the $i$th sample, and $h_j$ denotes negative sample features in the batch.

### 3.4 Unified Training Framework

**Overall objective.** Our training process combines both reconstruction and contrastive learning objectives to enhance high-quality image generation while simultaneously learning robust features for discriminative tasks. The overall training loss is formulated as Equation (7), where $\lambda$ is a reweighting parameter that balances the contributions of the reconstruction and contrastive objectives. We set $\lambda = 0.1$ for the training process, chosen via grid search as shown in Table 10a.

$$\mathcal{L} = \mathcal{L}_{\text{recon}} + \lambda \times \mathcal{L}_{\text{contrast}} \tag{7}$$

**Noise schedule in diffusion process.** Unlike standard sine or cosine noise schedules commonly used in diffusion model training, we adopt a modified schedule based on the observation that noise levels influence task suitability: low-level noise inputs benefit classification, while high-level noise inputs are more suited for generation. We used an inverse-cosine noise schedule (Hudson et al., 2024) to create more appropriate training samples for both objectives.

**Parameter-efficient training.** To maintain efficiency, we freeze all parameters of the pre-trained Stable Diffusion model and introduce trainable LoRA matrices within its cross-attention layers. These low-rank adaptation matrices enable fine-tuning while preserving the original model's weights, significantly reducing the number of trainable parameters and computational overhead. We employ default LoRA settings (Hu et al., 2021) for rank and learning rate to achieve an optimal balance between efficiency and performance without compromising generative capabilities.

## 4 Experiments

### 4.1 Experimental Settings

**Evaluation.** We evaluate CONTRADIFF, our unified framework, on both image understanding and genera-

| Hyperparameter | Value |
|---|---|
| optimizer | AdamW |
| learning rate | $1.5 \times 10^{-4}$ |
| momentum | 0.9 |
| weight decay | 0.05 |
| batch size | 512 |
| learning rate schedule | cosine decay |
| training epochs | 100 |
| warmup epochs | 5 |

Table 1: **Pretraining settings.**

| Hyperparameter | Value |
|---|---|
| optimizer | SGD |
| learning rate | 0.01 |
| momentum | 0.9 |
| weight decay | 0.05 |
| batch size | 256 |
| learning rate schedule | cosine decay |
| training epochs | 50 |
| warmup epochs | 5 |

Table 2: **Linear probing settings.**

tion tasks. For understanding tasks, we use extracted features for linear probing on ImageNet-1K classification (Russakovsky et al., 2015) and report top-1 accuracy. We also examine cross-dataset generalization on CIFAR-100 (Krizhevsky et al., 2009) via few-shot transfer learning, and additionally report *zero-shot* kNN classification results. Finally, to assess spatial understanding, we include a visual correspondence evaluation on SPair-71k (Min et al., 2019), following Tang et al. (2023). For generation tasks, we assess unconditional and class-conditional image generation performance on ImageNet-256 and free-form text-to-image generation on MSCOCO (Lin et al., 2014).

**Hyperparameter settings.** Tables 1 and 2 summarize the hyperparameter settings used in the ContraDiff framework. Pretraining uses AdamW optimizer (Loshchilov, 2017) with a base learning rate of $1.5 \times 10^{-4}$, momentum of 0.9, and weight decay of 0.05. The batch size is set to 512, and training follows a cosine decay learning rate schedule over 100 epochs, with 5 warmup epochs. Linear probing applies SGD optimizer with a learning rate of 0.01, momentum of 0.9, and weight decay of 0.05. The batch size is 256, using a cosine decay learning rate schedule for 50 training epochs, including 5 warmup epochs. Image generation employs classifier-free guidance (CFG) with 100 diffusion steps as the default setting.

**Training details.** We adopt pre-trained Stable Diffusion v1.4 as the base model with LoRA matrices attached to its cross-attention layers. We chose Stable Diffusion version 1.4 instead of stronger versions for fair comparison with other baselines, demonstrating that our method does not rely solely on heavily pretrained models. We trained ContraDiff on ImageNet-1K dataset. We used features from the bottleneck layer of UNet in Stable Diffusion, processed through cross-attention for downstream classification tasks. We directly used the diffusion model output for the image generation task. Our experiments were conducted on 4 Nvidia H100 GPUs. We trained ContraDiff for 100 epochs using a batch size of 512 with standard image augmentation techniques.

**Latent encoder and decoder.** These components encode input images into a compact latent space and decode them back into images. Leveraging this compressed latent space reduces computational overhead while facilitating efficient feature extraction. We applied the pre-trained VAE used in Stable Diffusion (Rombach et al., 2022) with a down-sampling factor $f = 4$ as our default encoder/decoder.

**Conditional encoder.** ContraDiff incorporates a conditional mechanism based on image inputs, encoding them into conditioning tokens. Similar to Rombach et al. (2022), we used a transformer-based conditional encoder with an embedding dimension of 512. However, instead of directly applying the pre-trained conditional encoder from Rombach et al. (2022), we adapted it to address the modality difference between text prompts and image inputs. The resulting embeddings are integrated into the denoising UNet through cross-attention layers, facilitating effective conditioning during the generation process.

**LoRA weight matrices.** LoRA (Hu et al., 2021) matrices efficiently adapt large pre-trained models by introducing trainable low-rank matrices to specific layers. In our implementation, we apply LoRA matrices to the cross-attention layers, allowing the model to tailor its responses to inputs from different modalities with minimal added parameters. This approach preserves the core features learned by the pre-trained Stable

| Method | Backbone | #Params. | | Acc.↑ |
| --- | --- | --- | --- | --- |
| | | **Trainable** | **Frozen** | |
| *contrastive based methods* | | | | |
| SimCLR (Chen et al., 2020) | ResNet50×2 | 94M | - | 74.1 |
| DINO (Caron et al., 2021) | ViT-B/16 | 86M | - | 78.0 |
| iBOT (Zhou et al., 2021) | ViT-B/16 | 86M | - | 75.8 |
| *generative based methods* | | | | |
| MAE (He et al., 2022) | ViT-L/16 | 304M | - | 73.5 |
| MAGE (Li et al., 2023b) | ViT-L/16 | 304M | 24M | 78.9 |
| GIVT† (Tschannen et al., 2025) | ViT-L/16 | 304M | - | 65.1 |
| *diffusion based methods* | | | | |
| DifFeed (Mukhopadhyay et al., 2023) | UNet* | 31M | 554M | 76.8 |
| SD Features | UNet* | - | 980M | 71.8 |
| **ContraDiff** (ours) | UNet* | 68M | 980M | **80.1** |

Table 3: **Linear probing performance on ImageNet-1K.** We group all evaluated methods into 3 categories: contrastive based methods, generative based methods, and diffusion based methods. We directly extract features from pre-trained Stable Diffusion v1.4 model and evaluate the raw features' performance as a baseline (shown as SD Features in table). In ContraDiff, trainable parameters refer to LoRA matrices and feature extraction module.† means results are from original works; * means UNet architecture from pre-trained diffusion models; SD (Stable Diffusion)

Diffusion model while optimizing performance for new tasks. For our setup, we applied LoRA matrices with a rank of 16.

## 4.2 Evaluation Results

### 4.2.1 Image Classification

**Setup.** For linear probing, we attach a linear classifier to the features extracted from our frozen pre-trained models. The classifier is trained using SGD with a momentum of 0.9, a fixed learning rate of 0.01, and an $L_2$ regularization penalty. The linear classifier is trained on ImageNet for 50 epochs with a batch size of 256, using 20 denoising steps.

**Results.** Classification performance is evaluated with top-1 accuracy on the ImageNet validation set. As summarized in Table 3, ContraDiff outperforms the diffusion-based DifFeed (Mukhopadhyay et al., 2023) by 3.3%. Furthermore, ContraDiff surpasses contrastive-based and other generative-based methods while maintaining a significantly lower number of trainable parameters. For example, while DINO (Caron et al., 2021) and MAE (He et al., 2022) require 86M / 304M trainable parameters from their ViT backbones, ContraDiff achieves superior classification performance with 68M trainable parameters.

### 4.2.2 Visual Correspondence

Visual correspondence is a critical image understanding task used for 3D reconstruction, tracking, and segmentation. We evaluate features extracted from ContraDiff on semantic correspondence task to demonstrate its potential in more complex vision understanding tasks. In particular, features extracted from ContraDiff yield better keypoint matching on SPair-71k (Min et al., 2019) than the base pretrained diffusion model and other representation learning baselines, indicating stronger spatially grounded semantics.

### 4.2.3 Image Generation

**Setup.** We evaluate our model's generative capacity through the challenging tasks of unconditional / class-conditional image generation on ImageNet. After pretraining, no additional fine-tuning is applied for image

| Method | PCK@bbox ↑ |
|---|---|
| DINO | 33.9 |
| OpenCLIP | 38.4 |
| DIFT$_{sd}$ | 52.9 |
| **ContraDiff** (ours) | **53.0** |

Table 4: **PCK per image on SPair-71k.** All methods are not fine-tuned on this task. Diffusion based methods (DIFT, ContraDiff) outperforms other baselines by a clear margin.

| Method | Resolution | FID ↓ | IS ↑ |
|---|---|---|---|
| ICGAN[†] (Casanova et al., 2021) | 256 | 15.6 | 59.0 |
| ADM[†] (Dhariwal & Nichol, 2021) | 256 | 26.21 | 39.70 |
| GIVT[†] (Tschannen et al., 2025) | 256 | 11.02 | - |
| MAGE (ViT-L) | 256 | 7.04 | 123.5 |
| **ContraDiff** (ours) | 256 | **5.56** | **142.3** |

Table 5: **Unconditional ImageNet-256 generation performance (FID).** Results are obtained from computing FID between ImageNet validation set and model generated images at resolution $256 \times 256$. † means results are from original works. ContraDiff consistently outperforms all generative baselines.

| Method | Type | FID ↓ | IS ↑ |
|---|---|---|---|
| MaskGIT (Chang et al., 2022) | MIM | 6.18 | 182.1 |
| MAGE(ViT-B) (Li et al., 2023b) | MIM | 6.93 | **195.8** |
| ADM (Dhariwal & Nichol, 2021) | Diffusion | 10.94 | 101.0 |
| LDM (Rombach et al., 2022) | Diffusion | 10.56 | 103.5 |
| **ContraDiff** (ours) | Diffusion | **5.16** | 189.7 |

Table 6: **Class-conditional ImageNet generation performance.** The best performance is marked bold and the second best result is underlined.

| Method | Trained on MSCOCO | FID↓ | CLIP Score↑ |
|---|---|---|---|
| U-Net (Ronneberger et al., 2015) | ✓ | 18.73 | 79.41 |
| LDM (Rombach et al., 2022) | ✗ | 23.31 | 84.65 |
| SD v1.4 | ✗ | 20.52 | 88.10 |
| **ContraDiff** (ours) | ✗ | **16.37** | **92.45** |

Table 7: **MSCOCO-256 Text-to-Image Generation.** FID computed from 40K samples is reported.

generation. The quality of the generated images is evaluated using Inception Score (IS) and Fréchet Inception Distance (FID). We generate 50k images at resolution 256×256, using 100 denoising steps per image, and calculate the metrics on the ImageNet-256 validation set.

**Unconditional image generation.** ContraDiff achieves an FID of 5.56 and an IS of 142.3 on unconditional ImageNet-256 generation, indicating strong image quality and diversity. Comparative results with other state-of-the-art models are provided in Table 5. These results demonstrate ContraDiff's ability to generate diverse, high-quality images without relying on additional labeled data. This success indicates the potential of large pre-trained diffusion models in applications requiring detailed and varied image synthesis, especially in scenarios where explicit class labels are unavailable.

**Class-conditional image generation.** ContraDiff inherently supports class-conditional image generation, leveraging its text-to-image diffusion model capabilities. For direct class-label conditioned generation, we adopt a conditional encoder similar to Rombach et al. (2022) consisting of a single learnable embedding layer

| Method | Type | Backbone | #Params. | Acc.@25↑ | Acc.@0↑ |
|---|---|---|---|---|---|
| SimCLR (Chen et al., 2020) | Contrastive | ResNet50×2 | 94M/- | 58.9 | 52.3 |
| MAGE (Li et al., 2023b) | Generative | ViT-L/16 | 304M/24M | 72.0 | 63.5 |
| DifFeed (Mukhopadhyay et al., 2023) | Diffusion | UNet* | 31M/554M | 70.3 | 61.8 |
| **ContraDiff** (ours) | Diffusion | UNet* | 68M/980M | **73.1** | **65.2** |

Table 8: **Transfer learning performance on CIFAR-100.** Top-1 accuracy of transfer learning on CIFAR-100 dataset of models pretrained on ImageNet-1K is reported. We choose one baseline method from each of our three groups of methods listed in Table 3 in the main paper. ContraDiff maintains the best performance over three baselines. We present the number of both trainable/frozen parameters in **"#Params."** column. * means UNet architecture from pre-trained diffusion models.

with a dimensionality of 512. We assess ContraDiff's conditional generation performance on the ImageNet-1K validation set and compare it against baseline methods, with results summarized in Table 6. ContraDiff achieves a significantly improved FID score of 5.16, indicating superior image quality and diversity compared to baseline models. Furthermore, it attains a high IS of 189.7, closely matching the top-performing MAGE model (195.8), demonstrating its effectiveness in class-conditional generation. The slight difference in the IS score may stem from ContraDiff's pretraining on large-scale datasets with distributions differing from ImageNet, which is used for IS evaluation. This distribution mismatch can influence the IS score while still preserving a high FID score. Overall, these results underline ContraDiff's effectiveness in balancing image fidelity and semantic alignment.

**Text-to-image generation.** We evaluate text-to-image generation using MSCOCO captions and standard CLIP-based metrics alongside FID, as shown in Table 7. We apply text encoder from SD v1.4 for text context and add image context from image encoder output from Gaussian noise input. These results verify that our joint objectives and LoRA adaptation do not degrade free-form prompt image generation: ContraDiff achieves 92.45 CLIP score, compared to 88.10 for the SD v1.4 baseline, indicating improved prompt adherence.

### 4.2.4 Transfer Learning on Classification

**Few-shot learning.** To evaluate the generalization ability of ContraDiff, we measure its performance on the CIFAR-100 dataset under a low-data regime, where only 25 samples per class are used for training. As shown in Table 8, ContraDiff outperforms all selected baseline methods, demonstrating its robustness in low-data regimes. These results highlight its capacity to extract meaningful representations and maintain strong performance even with limited training data.

**Zero-shot learning.** To further isolate representation quality from supervised adaptation, we additionally evaluate **zero-shot** performance on CIFAR-100 with kNN classification. As shown in Table 8, ContraDiff outperforms baselines on zero-shot kNN accuracy, consistent with the few-shot results, indicating that the learned diffusion representations generalize beyond ImageNet fine-tuning and remain effective on a different domain.

### 4.3 Ablation Study

**Ablation of individual components in ContraDiff.** Table 9 illustrates the contribution of each component to the performance of ContraDiff, starting from the pre-trained Stable Diffusion v1.4 baseline, which achieves 71.8% accuracy using the direct bottleneck layer output for linear probing. i) Adding an attention-based feature extraction network improves accuracy to 74.3% (+2.5%). ii) Incorporating LoRA training further boosts accuracy to 78.0% (+3.7%), with only the reconstruction objective applied. iii) Finally, adding a contrastive loss achieves an accuracy of 80.1% (+2.1%). Overall, ContraDiff demonstrates an 8.3% improvement over baseline (SD v1.4), with LoRA and contrastive loss providing a significant boost for optimal performance. We additionally report a concise breakdown for inference latency caused by each component.

| Component | Inference Latency | Acc.↑ |
|---|---|---|
| SD v1.4 | 1.858±0.008 | 71.8 |
| + Feature Extraction | 1.861±0.011 (+0.1%) | 74.3 (+2.5) |
| + LoRA Training | 2.094±0.019 (+12.0%) | 78.0 (+3.7) |
| + $\mathcal{L}_{\text{contrast}}$ (CONTRADIFF) | 2.094±0.019 (+12.0%) | 80.1 (+2.1) |

Table 9: **Ablation of ContraDiff components on linear probing accuracy and inference latency.** All additional components provide an increase in performance. Contrastive loss significantly improves accuracy compared to only using the reconstruction objective in LoRA training, achieving an 8.3% gain over the SD-v1.4 baseline. Inference latency is based on generation and refers to seconds used to generate one image with 100 steps.

| $\lambda$ | FID↓ | Acc.↑ |
|---|---|---|
| 0 | 14.71 | 78.0 |
| $1e^{-3}$ | 12.32 | 79.7 |
| $1e^{-1}$ | **5.56** | **80.1** |
| 1 | 5.45 | 78.3 |

(a) Contrastive loss weight $\lambda$.

| $(t_1, t_2, t_3)$ | Acc.↑ |
|---|---|
| random | 70.7 |
| late | 68.8 |
| early | 65.3 |
| uniform | **80.1** |

(b) Feature extraction timesteps.

| noise schedule | FID↓ | Acc.↑ |
|---|---|---|
| linear | 11.02 | 62.5 |
| cosine | 15.83 | 68.2 |
| sine | 17.49 | 70.4 |
| inverse-cosine | **5.56** | **80.1** |

(c) Noise schedule for training.

Table 10: **Ablation study on design choices.** We evaluate top-1 accuracy of extracted feature with kNN classification and FID of generated images on ImageNet-1K. Default hyperparameters are marked bold.

**Impact of contrastive loss via weighting parameter $\lambda$.** Our ablation study examines the influence of the contrastive loss during training, as shown in Table 10a. Experiment shows that $\lambda = 0.1$ provides the best overall performances on both tasks. We noticed that increasing $\lambda$ does not always lead to improved linear probing accuracy, which supports our unified framework: the combined loss benefits both tasks. The reconstruction loss acts as a regularizer for the classification task, meaning that increasing $\lambda$ may not necessarily improve linear probing performance. Therefore, our default $\lambda$ is chosen to balance performance across both tasks.

## 4.4 Discussion

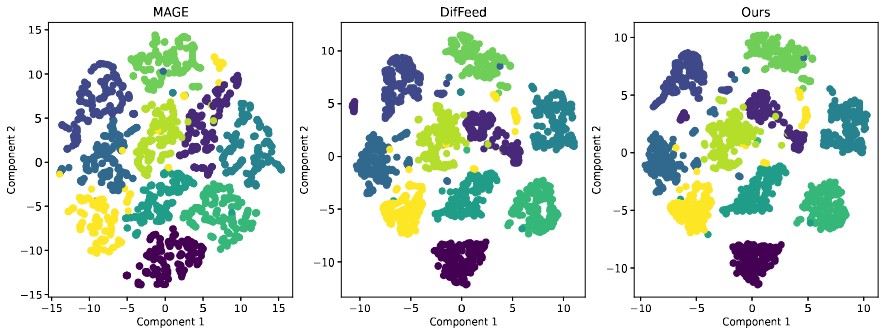

Figure 3: **ContraDiff produces more separable feature embeddings.** By t-SNE visualization on feature linear separability, we compare ContraDiff *(unified method)* against MAGE (Li et al., 2023b) *(unified method)*, which also uses a contrastive-based loss, and DifFeed (Mukhopadhyay et al., 2023) *(generative method)*, which elucidates feature extraction method design on a frozen diffusion model.

**ContraDiff's feature representations.** We use t-SNE to visualize feature vectors from the ImageNet validation set extracted by MAGE, DifFeed, and our method, as shown in Figure 3. We observed that

| Model | Total Params. | Trainable Params. |
|---|---|---|
| SimCLR ([Chen et al., 2020](#)) | 94M | 94M |
| MAGE ([Li et al., 2023b](#)) | 328M | 304M |
| DiffFeed ([Mukhopadhyay et al., 2023](#)) | 585M | **31M** |
| **ContraDiff** (ours) | 1048M | 68M |

Table 11: **Trainable vs. total parameters across methods.** ContraDiff only has more trainable parameters than DiffFeed ([Mukhopadhyay et al., 2023](#)), whereas the latter only makes use of frozen diffusion features and does not show competitive performance.

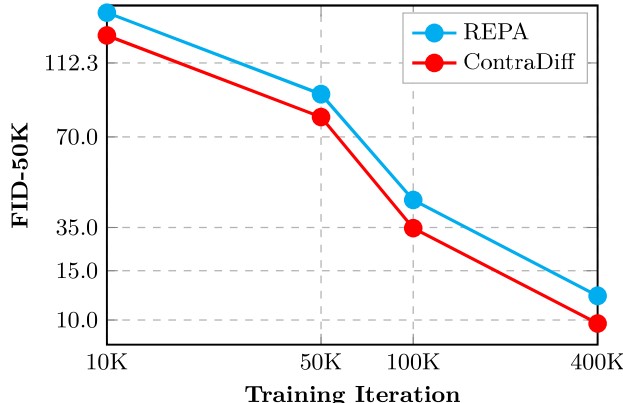

Figure 4: **Training convergence and efficiency.** ContraDiff achieves more rapid training convergence compared with REPA, a representation alignment method using external vision encoder representations.

our method produces more distinct and well-separated clusters, affirming the discriminative strength of our model's features.

**Efficiency analysis.** To improve computational efficiency, our model minimizes the number of trainable parameters while maintaining competitive performance. Table 11 compares the total number of trainable parameters across different models. As demonstrated, ContraDiff reduces the number of trainable parameters by 28% compared to SimCLR and 78% compared to MAGE. In Fig. 4, we report FID convergence during training iteration compared with REPA when trained from scratch on a SiT model. ContraDiff consistently achieves lower FID-50K than REPA at during 10K–400K iterations, with a clear gap already visible in early training.

## 5 Conclusion

In this paper, we introduced ContraDiff, a novel framework that efficiently adapts pretrained diffusion models for both generative and discriminative tasks within a unified framework. By combining reconstruction and contrastive losses and utilizing varying noise levels to balance the demands of both tasks, ContraDiff demonstrates strong performance and enhanced computational efficiency. The integration of LoRA within the cross-attention layers of Stable Diffusion models significantly reduces the number of trainable parameters, making the framework more resource-efficient without sacrificing accuracy. Our extensive evaluation highlights the framework's potential to address critical challenges in the field of self-supervised learning and generative modeling, such as fast adaptation of pretrained diffusion models to a variety of discriminative tasks. A promising direction for future research would be the extension of ContraDiff to integrate additional tasks and modalities.

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

## Supplementary Material Overview

This supplementary material provides additional insights into our method, including detailed implementation specifications and visualization. For implementation, we visualize the inference pipeline of CONTRADIFF. Visualization presents additional visualizations on generation results.

## A    Inference Pipeline

As shown in Figure 5, for the classification task, the feature map from the attention head predicts class labels. For generation, this feature guides image synthesis based on conditions. For unconditional generation, a pure Gaussian noise image ( $T = T_{max}$) is used as input.

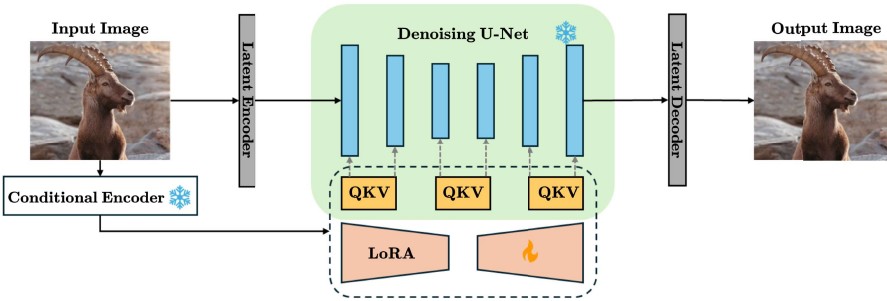

Figure 5: **Unified inference for classification and generation.** The condition encoder processes an input image to produce a feature representation, which serves as conditional latent for the denoising UNet. In the classification task, the feature map output from the attention head is used to predict class labels. For the generation task, this conditional latent guides the synthesis of coherent images according to the given input conditions. For unconditional image generation, a Gaussian noise image is used as conditional input.

## B    Visualization

We show some of the generated results by ContraDiff and compare them with outputs from pre-trained MAGE ViT-B/16 model, as illustrated in Figure 6.

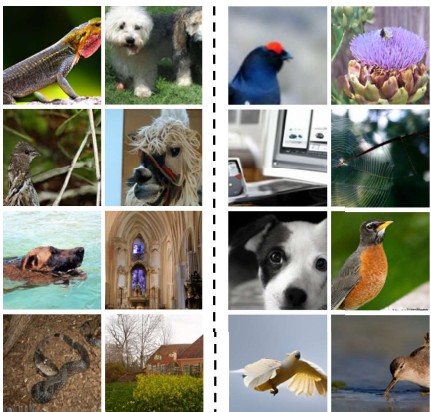

Figure 6: **ContraDiff improves ImageNet sample quality.** Generated images from MAGE (Li et al., 2023b) pre-trained ViT-B/16 model (left) and CONTRADIFF (right). We employ unconditional generation on ImageNet. ContraDiff brings images with more vivid details, illustrating its strong performance on generating high-fidelity images.

