# OpenReview forum: "ContraDiff: Unifying Training Process of Generative and Discriminative Vision Tasks in One Diffusion Model"
_TMLR — Rejected by TMLR_

### Review · Reviewer_C7tJ · 2026-01-23

**Summary Of Contributions:**

This work aims to unify the generation task and the discriminative task (e.g. classification) in one framework. The key idea is to take a pretrained diffusion model that can already generate images as a backbone, to train with standard diffusion loss. Then, extract different timestep levels of features from it to train through a contrastive target for discrimination. Combining these two losses together, the authors get a unified framework. Experiments are conducted to evaluate the proposed framework.

Strength:
1. Well-formatted and easy to follow.
2. The motivation to unify two tasks is clear and interesting.
3. The proposed framework is straightforward and effective.
4. Performance on both tasks achieves better results compared to baseline works.

Weakness:
1. Lack more details on how to select timesteps for feature extraction t1,t2,t3. Only mentioning a reference work is not enough.  And lack of an ablation study to validate this strategy.
2. I think here we lack one important analysis, whether the generation loss and contrastive loss could boost each other. What if we simply fine-tune with the generation loss, will the model perform better on generation? What if we focus on classification only and only use contrastive loss, will it perform better on classification? Because, based on my understanding, a unified model might perform worse than the model we trained with a single loss to focus on a single task. I encourage the authors to provide such an experiment and discuss more insights in this work.
3. This is a minor point.  I feel like the comparison is a bit unfair because this work takes a pretrained diffusion model as a backbone, but other baseline works are training from scratch.

Overall, I think it is an interesting paper and should be accepted after adding key experiments.

**Audience:**

Yes

**Audience Explanation:**

The topic is interesting for readers.

**Claims And Evidence:**

Yes

**Claims Explanation:**

See above.

**Requested Changes:**

See weakness 1.2.

---

> ### Author Response · Authors · 2026-02-20
> **Response to reviewer C7tJ**
>
> We thank the reviewer for comments that help us improve the ablation and analysis of the proposed framework.
>
> - Weakness 1 - need details/ablation for timestep selection $(t_1,t_2,t_3)$:
> We added an explicit ablation on timestep selection strategies (Table 10b). Uniform sampling over timesteps achieves the best accuracy; early- or late-biased selections degrade performance. This supports our default multi-timestep feature design.
>
> - Weakness 2 - do generation loss and contrastive loss boost each other: We acknowledge that single-task training can outperform unified training. Meanwhile, we are studying the dynamics in model performance when the balance between generative loss and contrastive loss varies: Table 10a directly isolates the effect of the contrastive term by sweeping its weight $\lambda$ while keeping the rest of the pipeline identical. When $\lambda=0$ (i.e., generation-only fine-tuning), generation quality is worse (FID 14.71) and the discriminative performance is also lower (Acc. 78.0). Introducing a moderate contrastive loss weight improves both objectives simultaneously: at $\lambda=0.1$, we obtain the best joint performance with FID 5.56 and Acc. 80.1. This behavior is consistent with the two losses being complementary: the contrastive term regularizes/structures the representation learned during diffusion adaptation, improving linear separability while also improving image generation quality. This finding is aligned with previous observations in [1]. We consider a more comprehensive study on the trade-off as future work.
>
> - Weakness 3 - comparison may be unfair:
> We acknowledge the concern and as a preliminary study, we evaluated image generation FID on a training-from-scratch SiT with ContraDiff loss and compared with REPA. The training curve in Figure 4 shows that ContraDiff yields faster convergence on image generation performance.
>
> [1] Yu, Sihyun, et al. "Representation alignment for generation: Training diffusion transformers is easier than you think." arXiv preprint arXiv:2410.06940 (2024).

---

### Review · Reviewer_TAoV · 2026-02-04

**Summary Of Contributions:**

The paper proposes "ContraDiff," a method aimed at unifying generative and discriminative vision tasks within a single diffusion model. The authors build upon a pre-trained Stable Diffusion (v1.4) backbone and introduce a fine-tuning strategy that incorporates Low-Rank Adaptation (LoRA) and a contrastive loss function. Specifically, the contrastive objective treats intermediate states at different noise levels as positive views to enhance feature separability. The authors claim this approach achieves competitive performance on ImageNet classification (linear probing) and generation (FID) compared to baselines like MAGE and DiT.

The authors position the work as proposing a "unified framework" that solves the computational inefficiency of previous methods (e.g., MAGE). However, I am suspicious of this framing because:
- Prior Exploration: The conceptual combination of diffusion (generative) and contrastive (discriminative) objectives has already been explored in prior art like MAGE (Li et al., 2023b), which successfully learns joint representations via masking and contrastive loss. The proposed loss design in this paper is straightforward and I hope the author clarify what is the contribution more concretely.
- Source of Efficiency: The paper argues the framework achieves "computational efficiency." However, this efficiency does not appear to stem from the architecture or the unified framework itself, but rather from using LoRA.

**Audience:**

Yes

**Audience Explanation:**

practitioners working with Foundation Models and Efficient Transfer Learning.

**Claims And Evidence:**

No

**Claims Explanation:**

1. The authors are motivated by the question: "Can we develop a unified framework that... is computationally efficient?" They compare their training cost favorably against MAGE (which requires 1600 epochs). This comparison is unfair and the evidence provided seems misleading.
- MAGE is a pre-training method that learns from scratch (or from raw pixels).
- ContraDiff utilizes Stable Diffusion v1.4, which has already undergone massive-scale pre-training on LAION-5B.
The "efficiency" observed is a direct result of utilizing pre-trained weights and the LoRA optimization technique, not an inherent property of the proposed "ContraDiff framework." If one were to fine-tune MAGE from a similar pre-trained checkpoint using comparable parameter-efficient techniques, the efficiency gap might disappear. The authors claim the framework is efficient, but the evidence only proves that fine-tuning via LoRA is efficient. The authors must explicitly decouple the benefits of the initialization/optimization (SD + LoRA) from the benefits of their proposed method.

2. Clarity of Contribution regarding the Loss Function:
The paper claims to propose a unified framework, but the core methodological component -- the contrastive loss -- is standard. Given that MAGE has already established the efficacy of combining generative and contrastive objectives, the paper fails to justify what constitutes the specific contribution of this framework. Is the contribution simply the empirical observation that this specific loss works well with LoRA on Stable Diffusion? If so, the claims should be adjusted to reflect that this is an empirical adaptation strategy rather than a novel framework proposal. The current text implies a higher level of methodological innovation than the evidence supports.

**Requested Changes:**

please see the above concerns about contribution

---

> ### Author Response · Authors · 2026-02-20
> **Response to reviewer TAoV**
>
> We thank the reviewer for helping to clarify the contribution of our work and evaluate the framework efficiency.
>
> - Weakness 1 - Efficiency comparison might not be fair:
> We agree that directly comparing **ContraDiff** (training with LoRA) with training the entire model from scratch may not be the most convincing way to justify the benefits of our method. As shown in Figure. 4, we added an experiment where we train a vanilla SiT model from scratch using ContraDiff loss for 400K steps and report image generation performance. Results show that ContraDiff yields faster training convergence compared against [1], which relies on a powerful external visual representation for supervision.
> With that said, we agree that comparisons to methods trained from scratch must be framed carefully and we therefore revise our positioning: ContraDiff is a unified, parameter-efficient adaptation framework for pretrained diffusion backbones, rather than a claim of superior compute efficiency over from-scratch pretraining.
> To support efficiency within this adaptation regime, we provide
> (i) trainable vs total parameter counts (Table 11),
> (ii) measured inference latency (Table 9), and
> (iii) a training convergence comparison to [1] under matched iteration budgets (Figure 4), where ContraDiff reaches comparable FID in fewer iterations.
> Importantly, Table 9 decouples the sources of gains: LoRA (reconstruction-only) provides a large improvement, and adding the contrastive objective yields further accuracy gains without additional latency beyond LoRA, showing benefits beyond ''LoRA alone''.
>
> - Weakness 2 - specify the contribution in loss design:
> We agree the high-level idea of mixing generative and contrastive objectives has precedent.
> Our contribution is diffusion-specific and pretrained-adaptation-focused: we treat diffusion noise levels as views and apply contrastive learning within the denoising process, while using parameter-efficient LoRA updates to retain/boost generation and learn discriminative representations in one pipeline (Figure 2). This is validated across ImageNet linear probing (Table 3), ImageNet unconditional/class-conditional generation (Tables 5-6), MSCOCO text-to-image (Table 7), CIFAR-100 transfer/zero-shot (Table 8), and SPair-71k correspondence (Table 4), with targeted design ablations (Tables 9-10).
>
> [1] Yu, Sihyun, et al. "Representation alignment for generation: Training diffusion transformers is easier than you think." arXiv preprint arXiv:2410.06940 (2024).

---

### Review · Reviewer_aVCN · 2026-02-05

**Summary Of Contributions:**

This paper introduces ContraDiff, a unified and parameter-efficient framework that adapts pretrained diffusion models to perform both generative and discriminative vision tasks within a single training pipeline. The core idea is to leverage intermediate denoising representations from diffusion models and jointly optimize them using a reconstruction loss and a contrastive loss across multiple noise levels.

To improve efficiency and flexibility, the method applies LoRA-based parameter-efficient fine-tuning to a pretrained Stable Diffusion model, substantially reducing the number of trainable parameters and training cost while preserving performance. Extensive experiments on ImageNet-1K demonstrate that ContraDiff achieves competitive classification accuracy (80.1%) alongside high-quality unconditional image generation (FID 5.56 at 256×256). The results show that contrastive representation learning and diffusion-based generation can coexist and mutually benefit each other in a single framework.

Key Strengths
1. The combination of reconstruction and contrastive losses across varying noise levels is intuitive and well aligned with the diffusion process.
2. Exploits denoising steps as rich intermediate representations, which are often underutilized in diffusion models for discriminative tasks.
3. Achieves competitive ImageNet-1K classification accuracy while maintaining high image generation quality, outperforming some task-specific contrastive methods.

Key Weaknesses
1. Experiments focus primarily on ImageNet-1K classification and unconditional image generation; additional understanding tasks (e.g., captioning, VQAm detection, segmentation) would strengthen the claims.
2. The framework is evaluated only on unconditional and class-conditioned image generation; no free-from text conditioned image generation was evaluated, leaving the question of if there are any free-from text condition generation regression caused by the joint generation and understanding training.
3. While parameter-efficient, the approach still assumes access to powerful pretrained diffusion models, which may limit accessibility.
4. More ablation or theoretical analysis on why specific noise levels or loss balancing strategies work best could further clarify the method’s behavior.
5. The authors strengthened the training efficiency brought by LoRA fine-tuning, however, the authors have not analyzed inference latency.

**Audience:**

Yes

**Audience Explanation:**

Even with the identified empirical gaps, the findings of this paper would be of decent interest to several key subgroups:

1. There is an active and growing interest in "generalist" vision models. Researchers in this space would be interested in the Noisy Contrastive Learning mechanism as a potential strategy for aligning generative and discriminative features within a single backbone.
2. The finding that denoising intermediates can be repurposed for high-accuracy classification (80.1% on ImageNet) provides a new perspective on the latent representations learned by diffusion models, moving beyond their traditional role as "pure" generators.
3. Those working on LoRA and parameter-efficient fine-tuning would find the methodology valuable for understanding how to steer massive pre-trained models (like Stable Diffusion) toward non-native tasks without a total retraining.

**Claims And Evidence:**

No

**Claims Explanation:**

While the reported results on ImageNet-1K are empirically accurate, the evidence is not fully convincing due to the significant gap between the paper's broad claims of "unifying vision tasks" and its narrow experimental scope. The lack of evaluation for free-form text-to-image generation and downstream understanding tasks (e.g., segmentation, zero-shot CIFAR-100) leaves the risks of generative regression and poor generalization unaddressed. Furthermore, by emphasizing parameter efficiency through LoRA while omitting an analysis of inference latency, the authors fail to provide a clear picture of the model's practical utility compared to standard discriminative backbones.

**Requested Changes:**

1. The model was trained on ImageNet labels. Authors need to prove it hasn't lost its ability to follow complex text prompts (like "a cat riding a bike in space"). Show CLIP Scores or MS-COCO results to confirm the creative side hasn't regressed.
2. Authors claim the model is "efficient" because it has fewer parameters, but diffusion models are notoriously slow at "thinking" (inference). 3. Authors should provide a table comparing inference latency to baseline models.
3. Doing well on ImageNet is one thing, but can the model recognize objects it wasn't specifically fine-tuned on? Provide Zero-Shot scores on CIFAR-100 to prove the model's "brain" is actually getting smarter, not just memorizing a specific dataset.
4. If this is truly a "unified" vision model, show us it can do more than just pick a label. Can it generalize to other more complex understanding tasks?
5. Authors use different noise levels for training. Which noise levels are actually the most important for learning? A simple chart showing this would make the science much clearer.

---

> ### Author Response · Authors · 2026-02-20
> **Response to reviewer aVCN**
>
> We thank the reviewer for the comments and questions which will help improve the solidity of the submission.
>
> - Weakness 1 - more understanding tasks:
> We add a semantic correspondence benchmark on SPair-71k (Table 4). Although ContraDiff is not trained specifically for correspondence, it exhibits strong keypoint matching performance matching DIFT [1], indicating improved spatial semantics that are foundational for downstream tasks. In addition, we evaluate generalisation ability on CIFAR-100 and report zero-shot kNN accuracy (Table 8), which better reflects representation quality without task-specific fine-tuning.
>
> - Weakness 2 - free-form text-conditioned image generation:
> We add free-form text-to-image generation evaluation on MSCOCO-256 and report FID and CLIP score (Table 7). ContraDiff improves both metrics over SD v1.4, suggesting the joint objectives do not regress text-conditioned generation quality; the CLIP-based score also indicates improved text-image alignment under this metric.
>
> - Weakness 3 - access to powerful pretrained diffusion models:
> To address concerns about reliance on pretrained diffusion models, we add a preliminary training-from-scratch experiment with a SiT backbone and compare with REPA [2]. The results (Figure 4) show improved convergence behavior under matched iteration budgets, suggesting promise for training-from-scratch settings.
>
> - Weakness 4 - ablation on noise levels:
> We added an ablation on feature extraction timesteps (t1, t2, t3) (Table 10b). The results show uniformly spaced timesteps perform best, while ``early/late'' biased selections underperform. We also include a noise schedule ablation (Table 10c) and report the effect of varying the contrastive weight $\lambda$ (Table 10a), clarifying how noise-level choices and loss balancing affect performance.
>
> - Weakness 5 - inference latency analysis:
> Table 9 reports component ablations with both linear probing accuracy and latency. In the revised version, we report inference latency as an extra column in Table 9. We evaluate inference latency as seconds per image with 100 denoising steps. Feature extraction adds negligible overhead ($0.1\%$), and LoRA adds $12\%$ latency with a clear margin in performance improvement. Contrastive loss additions do not further increase latency beyond LoRA’s cost. While we acknowledge diffusion inference is slower than discriminative encoders, we limit the latency comparison within our method ablation to quantify the incremental overhead of ContraDiff relative to the SD baseline during generation, rather than to claim diffusion is faster than discriminative encoders.
>
> [1] Tang, Luming, et al. "Emergent correspondence from image diffusion." Advances in neural information processing systems 36 (2023): 1363-1389.
>
> [2] Yu, Sihyun, et al. "Representation alignment for generation: Training diffusion transformers is easier than you think." arXiv preprint arXiv:2410.06940 (2024).

---

### Author Response · Authors · 2026-02-20
**General Response to Reviewers**

We thank the reviewers for their constructive feedback and questions. In the revision, we added new experiments and ablations to address the reviewers' concerns and improve our framing and claims. The core goal of the added experiments is to demonstrate **ContraDiff**'s ability on more diverse generation and visual correspondence tasks. We also conduct additional ablations to further justify the effectiveness of our training strategy. The summary of our changes is as follows:

- Expanded evaluation on generation and understanding tasks: We provide zero-shot kNN CIFAR-
100 classification results in addition to our existing few-shot experiment (Table 8). We provide
visual semantic correspondence results on SPair-71k following the DIFT setup (Table 4). We add
text-to-image generation evaluation on MSCOCO using FID and CLIP score (Table 7).

- Further ablations on representation extraction design and loss function: We add an ablation on
timestep selection in Table 10b and report the effect of varying the contrastive weight $\lambda$ to characterize the dynamics on reconstruction and contrastive objectives (Table 10a).

- Revised and clarified efficiency claims: We agree that the efficiency primarily comes from operating
in a pretrained setting and using LoRA. We therefore sharpen the claim: ContraDiff is a unified
adaptation pipeline that achieves strong generative and discriminative performance while
keeping trainable parameters small (Table 11) and providing measured inference latency (Table 9).
Additionally, the REPA convergence comparison (Fig. 4) shows improved convergence behavior under matched iteration budgets.

---

### Decision · Action_Editor_Bap3 · 2026-03-24

**Recommendation:** Reject

**Audience:**

Yes

**Audience Explanation:**

Yes, all reviewers agree.

**Claims And Evidence:**

No

**Claims Explanation:**

This paper proposes a method to fine-tune diffusion models to make them capable of performing discriminative tasks in addition to generation. Several reviewers were not convinced by the empirical evaluation of the method, with several issues being highlighted. In particular, reviewers were not satisfied with the extent of the considered discriminative tasks and the way the comparisons between fine-tuned foundation models were performed against end-to-end-trained baselines. Reviewers also highlighted the "unifying" wording in the paper as too strong to be backed by evidence. In light of these issues, I recommend rejection.

**Resubmission Of Major Revision:**

The authors may consider submitting a major revision at a later time.